# Caries Risk Assessment Using the Caries Management by Risk Assessment (CAMBRA) Protocol among the General Population of Sakaka, Saudi Arabia—A Cross-Sectional Study

**DOI:** 10.3390/ijerph19031215

**Published:** 2022-01-22

**Authors:** Azhar Iqbal, Osama Khattak, Farooq Ahmad Chaudhary, Meshal Aber Al Onazi, Hmoud Ali Algarni, Thani AlSharari, Abdullah Alshehri, Mohammed Mustafa, Rakhi Issrani, Ebtehal Yanallah Mohamed Alghamdi, Alaa Yahya Ali Alghamdi, Nojoud Omar Ahmad Balubaid

**Affiliations:** 1Department of Opertative Dentistry and Endodontics, College of Dentistry, Jouf University, Sakaka 72388, Saudi Arabia; dr.osama.khattak@jodent.org (O.K.); dr.meshal.alonazi@jodent.org (M.A.A.O.); dr.hmoud.algarni@jodent.org (H.A.A.); 2Department of Community Dentistry, School of Dentistry (SOD), Federal Medical Teaching Institution (FMTI)/PIMS, Shaheed Zulfiqar Ali Bhutto Medical University (SZABMU), Islamabad 44000, Pakistan; 3Restorative and Dental Materials Department, Faculty of Dentistry, Taif University, Taif 26571, Saudi Arabia; Thani.Alsharari@gmail.com; 4Department of Conservative Dental Sciences, College of Dentistry, Prince Sattam Bin Abdulaziz University, Al-Kharj 11942, Saudi Arabia; am.alshehri@psau.edu.sa (A.A.); ma.mustafa@psau.edu.sa (M.M.); 5Department of Preventive Dentistry, College of Dentistry, Jouf University, Sakaka 72388, Saudi Arabia; dr.rakhi.issrani@jodent.org; 6General Dentists, Al Khobar City 34218, Saudi Arabia; DrEbtehaly_60@hotmail.com; 7General Dentists, Arar City 73211, Saudi Arabia; Alaa.y.g@hotmail.com; 8General Dentists, Al-Baha City 65511, Saudi Arabia; Dr.nojoud@hotmail.com

**Keywords:** caries risk assessment, CAMBRA, dental caries, disease indicators, protective factors

## Abstract

*Background*: Caries risk assessment is a useful tool in caries prevention and management. Using a tool such as CAMBRA, every individual can be assessed according to his or her disease indicators, risk factors, and protective factors for the current and future caries. *Aim*: This study aimed to assess caries risk among the general population of Sakaka, Saudi Arabia using the CAMBRA protocol. *Methods*: This cross-sectional study was conducted at university dental clinics using a questionnaire that was formulated using the CAMBRA caries risk assessment tool; afterwards, all 160 participants were intra-orally examined to assess oral hygiene status and presence of disease. Independent *t*-tests, ANOVAs, and chi-square tests were performed for analysis. Results: The majority of participants had one or more disease indicators, with white spots and visible cavities (71.3%), and the most commonly present risk factor was visible heavy plaque on teeth (82.5%). The use of fluoridated toothpaste (92.5%) was the most common protective factor. The majority of participants (85%) were in the ‘High’ category of Caries risk assessment. The prevalence of high caries risk was significantly higher among the rural participants compared to the urban (*p* <0.05), and significantly fewer of those with a primary school education level or lower were in the high dental caries risk group compared to the other educational categories (*p* <0.001). *Conclusion*: The caries risk among the general population of Sakaka, Saudi Arabia, is high, with significant variation among age groups, education levels, and geographical locations.

## 1. Introduction

Oral health diseases are a major global public health issue; in particular, dental caries is the most prevalent oral health disease, affecting 60–90% of children and adults worldwide [1]. Despite continuous advancement and development in science, dental caries continues to be a concern worldwide due to its multifactorial nature including the interaction among bacteria, diet, and host response [2]. A multi-dimensional preventive strategy is needed to control dental caries [2,3]. Individuals who are at risk of developing dental caries in the future must be identified and assessed using different tools and models. Risk assessment may be a useful tool in caries prevention and management. It can be used as a strategy for improving the efficiency and effectiveness of preventive procedures and programs. Better and more cost-effective treatment can be provided by employing risk assessment rather than providing treatments independent of the individual’s risk [2]. By assessing caries risk, the potential for patient care is greatly enhanced, as it is the cornerstone of a minimally invasive treatment plan allowing for the determination of the most appropriate invasive and non-invasive treatments and strategies for recall. Protective factors and caries indicators such as bacteria, absence of saliva, and poor dietary habits can be determined by caries risk assessment. Dental sealants, adequate salivary flow, antimicrobial therapies, fluoride use, and a controlled diet contribute towards maintaining healthy teeth and preventing dental caries [4]. The cariogram is a validated model used in many previous studies; however, its accuracy in pre-school children was found to be limited in several studies [5,6,7]. This limited accuracy raised the necessity for a new caries risk assessment system. Therefore, the Caries Management by Risk Assessment (CAMBRA) system was introduced nearly a decade ago to fill that gap.

The CAMBRA is an evidence-based risk assessment tool for the prevention and treatment of caries at the earliest stages, instead of waiting for irreversible damage to the teeth [8]. Using CAMBRA, every individual is assessed according to his or her disease indicators, risk factors, and protective factors to work out the risk for current and future caries [9]. It is the most commonly used and recommended tool for assessing caries risk in individuals aged six years through adulthood [10]. According to the findings from the literature, CAMBRA presents an efficient process for identifying individuals at high risk of dental caries who need preventive services and management of risk factors [8,9]. However, the CAMBRA protocol has never been used for assessing caries risk in the Saudi population. Hence, this study aimed to assess caries risk among the general population of Sakaka, Saudi Arabia using the CAMBRA protocol.

## 2. Materials and Methods

This cross-sectional study was conducted in the university dental clinics of the College of Dentistry, Jouf University, Sakaka, Saudi Arabia, using a Caries Risk Assessment (CAMBRA) protocol, from 15 March 2021 to 15 June 2021 [11]. Patients and their attendees (e.g., patients’ family members, relatives, and friends) visiting the outpatient departments of university dental clinics were selected. Patients who were six years old or more, understood English and Arabic, and were residents of Sakaka and its surroundings were included in the study. A simple random sampling technique was adopted for the current study. The monthly average number of patients attending the university dental clinics at the College of Dentistry, Jouf University, is approximately 600. The sample size of this study was calculated using this number as a guide; the response distribution was assumed to be 50% with 95% confidence levels and a 5% margin of error, which showed that a total of 160 subjects were needed. This study was approved by the ethical review board of Jouf University (Reference code: 25-06-42). The voluntary participation of all participants was ensured, and they were briefed regarding the purpose of the study before the research team obtained written informed consent, and in the case of minors, informed consent was taken from the parent/guardian. The participants were asked to fill out survey forms; afterwards they were intra-orally examined and a bitewing radiograph was performed by the research team for assessing their oral hygiene status and determining the presence of disease. This questionnaire was formulated using the CAMBRA caries risk assessment tool, including eight risk and protective factors and four disease indicators. Participants were categorized as low risk (no carious lesions, no plaque, optimal fluoride use, and regular dental care); moderate risk (carious lesion in previous 12 months, visible plaque, suboptimal fluoride, and irregular dental care); and high risk (one or more carious lesions, visible plaque, suboptimal fluoride, no dental care, high bacterial challenge, and inadequate saliva flow) accordingly [12]. The questionnaire was pilot-tested among 20 patients above six years of age, and was found to be reliable with a Cronbach’s alpha value above 0.75.

Data extraction sheets were used to collect data. Descriptive analysis (percentages, mean with standard deviation) was used to summarize the data. Age, gender, and other demographic characteristics were tested using the chi-square test wherever appropriate and inferential analyses (independent *t*-test, dependent *t*-test, and ANOVA). For correlation analysis, Pearson or Spearman correlations as per the type of data were used. All data were analyzed using version 24 of Statistical Package for the Social Sciences (SPSS IBM, Chicago, IL, USA).

## 3. Results

The proportion of males and females in the study population was comparable, and age-wise distribution showed that the most common age groups in the study population were between 20 and 49 years (81.4%). The majority of participants were from an urban area and had at least a secondary education (Table 1).

The majority of participants had one or more disease indicators, with white spots (71.3%) and visible cavities (70.0%) being the most common disease indicators. The most commonly present risk factors were visible heavy plaque on teeth (82.5%) followed by deep pits and fissures (68.8%), and the use of fluoridated toothpaste (92.5%) was the most common protective factor among participants (Table 2).

The majority of participants (85%) were found to be in the ‘High’ risk category and only 15% were in the ‘moderate’ category of caries risk assessment (Figure 1).

The age group of 20–29 years was found to have a significantly lower proportion of subjects with high dental caries risk compared to other age groups (*p* < 0.001). The prevalence of high caries risk was significantly greater among the rural population as compared to the urban living population (<0.05). Those with a primary school education level or lower had a significantly smaller proportion of subjects with high dental caries risk compared to other educational categories (<0.001) (Table 3).

## 4. Discussion

In this study, caries risk was assessed among the general population of Sakaka, Saudi Arabia, by employing CAMBRA. The comparable proportions of male and female subjects were noted in this study, with the majority being from the urban area and educated. This result conflicts with the study by Almusawi et al., where women were in the majority compared to men when identifying the caries risk using CAMBRA among diabetic patients [13]. Although ages from 6–>60 years were considered in this study, the majority of the subjects were aged between 20–49 years. This observation is similar to the study of Almusawi et al., which included diabetic subjects aged 30 years and above [13]. In the study by Qasim et al., they assessed the caries risk among the general population of Lahore, where they found a comparable distribution of the genders [14].

The findings of our study showed that over two-thirds (70%) of the subjects had either clinically or radiographically established dentin or enamel lesions, and over half of the subjects had had restorations in the past three years. These findings indicate that the general population is at risk of developing dental caries and early intervention may curtail severe sequelae. This will not only be psychologically beneficial but also economically profitable. The findings of our study are similar to other studies [10,15,16], where the majority of lesions were visible cavities and white spot lesions (85%) [10,13,15,16]. The observations of this study indicated the risk factors in the majority of the subjects (>80%) to be plaque (82%), deep pits and fissures (69%), and frequent food intake (67%). Categories including recreational drug use, orthodontic appliances, inadequate salivary flow, exposed roots and saliva reducing factors were less frequently seen compared to the above-mentioned risk factors. These observations are similar to the studies by Farsi N et al., Chaffee BW et al., and Almusawi et al., where similar risk factors were prevalent [9,13,16]. Previous studies have established the risk of caries and their relationship with plaque, deep pits and fissures, and frequency of food intake [17,18]. When associated with an enamel defect, plaque may exponentially increase caries risk [19,20]. The other risk factors that are calibrated in CAMBRA also aid in the calculation of the risk by optimally including as many risk factors as are commonly reported for caries.

The main protective factor in this study was fluoridated toothpaste, which most of the subjects used, and nearly half of the subjects had the habit of brushing twice. Less than tenth of the subjects applied fluoride mouth-rinse, fluoride varnish, or xylitol and calcium and phosphate paste. This study’s observations are comparable to the previous study by Featherstone et al., where they observed an increased number in the high-risk caries group irrespective of fluoride supplementation of the water or topical application, and suggested antibacterial agents to lower caries incidence [21]. In a clinical trial conducted by Featherstone et al., they concluded that fluoride therapy and targeted antibacterials significantly lowered the level of caries risk in the intervention groups [22].

The prevalence of dental caries is considerably lower in developed nations, which may be due to better living conditions, health awareness, application of fluorinated products, and preventive oral care programs [23,24,25]. Most of the participants in this study had a high risk of caries (85%) while the remainder were in the moderate-risk group. These findings were similar to many previous studies [12,14,26]; however, there are some studies where the moderate-risk group was in the majority [25,27]. The recruitment of participants in this study was from a dental department where participants mostly came for dental treatment, which might be a plausible reason for the high risk of caries among participants. In this study, only group II (20–29 years) had a greater proportion of subjects in the moderate-risk group and all the other age groups had most of the subjects in the high caries risk group. These findings are fairly similar to the studies by Qasim et al., where 43% of subjects were between 6–29 years of age and had moderate risk; however, contrary to our study, the majority of subjects in that study were female [14,15]. A significant number of participants with high caries risk were from rural areas in this study. This observation was unique, as while previous studies have focused on several aspects of CAMBRA, none of them have compared regional variations.

When education-wise comparison of caries risk was assessed in this study, moderate caries risk was only observed in the participants having a primary school or lower level of education, and the rest of the participants belonging to other education level groups had high caries risk. These results were in accordance with the study by Almusawi et al., where the majority of participants had more than a primary school level of education and had higher caries risk [13]. In another study on caries risk in children, the education of the parents was an important factor and significantly associated with a lower risk of caries among the children [28].

Accurate estimation of caries risk can help in improving patient education and interventions. Tools such as CAMBRA that are patient-centric and can be easily understood by the patient are essential for proper analysis of the risk. Once the factors that indicate risk are identified with the help of these tools, a personalized treatment plan can be designed for the patients. CAMBRA encourages patients in the decision-making process and can increase a good rapport with the dentist. A limitation of this study was the design of the study. This study included subjects attending the dental department, which may have altered the results, as the majority showed higher risk and there was no control group to compare the findings. Socioeconomic conditions, which have been shown to influence caries risk, were not measured in this study. Moreover, the study is self-reported, which results in inherent response bias.

## 5. Conclusions

Within the limitations of the study, the caries risk among the general population of Sakaka, Saudi Arabia, appears high, with significant variation among age groups, education levels, and geographical locations. Further studies are suggested with larger sample sizes and follow-ups to corroborate our findings.

## Figures and Tables

**Figure 1 ijerph-19-01215-f001:**
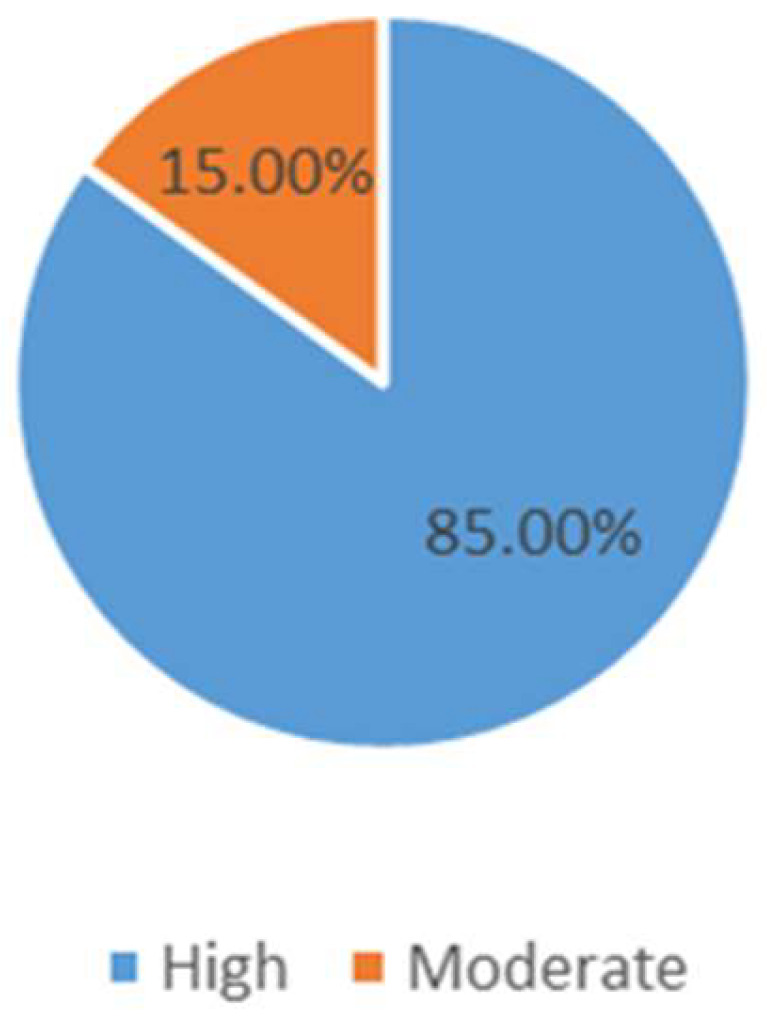
Distribution of study population according to caries risk assessment.

**Table 1 ijerph-19-01215-t001:** Socio-demographic characteristics of the participants.

Characteristics	Groups	Number (%)
Gender	Male	82 (51.2)
	Female	78 (48.8)
Age groups (years)	Group I (6 to 19 years)	0
	Group II (20 to 29 years)	42 (26.2)
	Group III (30 to 39 years)	42 (26.2)
	Group IV (40 to 49 years)	46 (28.8)
	Group V (50 to 59 years)	16 (10.0)
	Group VI (above 60 years)	14 (8.8)
Residence	Rural	51 (31.9)
	Urban	109 (68.1)
Education	Primary school and below = 1	45 (28.1)
	Secondary school = 2	54 (33.7)
	Bachelor = 3	35 (21.9)
	Diploma = 4	12 (7.5)
	Master = 5	10 (6.3)
	PhD = 6	4 (2.5)

**Table 2 ijerph-19-01215-t002:** Distribution of study participants according to disease indicators, risk, and protective factors.

Disease Indicators	*N* (%)
Visible cavities or radiographic penetration of the dentin	112 (70.0)
Radiographic approximal enamel lesions (not in dentin)	111 (69.4)
White spots on smooth surfaces	114 (71.3)
Restorations in last three years	89 (55.6)
**Risk Factors**	
Visible heavy plaque on teeth	132 (82.5)
Frequent snack (> 3 × daily between meals)	107 (66.9)
Deep pits and fissures	110 (68.8)
Recreational drug use	45 (28.1)
Inadequate saliva flow by observation	33 (20.6)
Saliva reducing factors (medications/radiation/systemic)	17 (10.6)
Exposed roots	30 (18.8)
Orthodontic appliances	43 (26.9)
**Protective Factors**	
Home/work/school is a fluoridated community	39 (24.4)
Fluoride toothpaste at least once daily	148 (92.5)
Fluoride toothpaste at least 2 × daily	68 (42.5)
Fluoride mouth rinse (0.05% NaF) daily	8 (5.0)
Fluoride varnish in last six months	12 (7.5)
Chlorhexidine prescribed/used one week each of last six months	38 (23.8)
Xylitol gum/lozenges 4 × daily last six months	14 (8.8)
Calcium and phosphate paste during last six months	4 (2.5)

**Table 3 ijerph-19-01215-t003:** Socio-demographic characteristics-wise comparison of caries risk.

		Caries Risk		*p*-Value
		High	Moderate	
		*N* (%)	*N* (%)	
Age group	6 to 19 years	22 (52.4)	20 (47.6)	<0.001
	20 to 29 years	39 (92.9)	3 (7.1)	
	30 to 39 years	45 (97.8)	1 (2.2)	
	40 to 49 years	16 (100.0)	0	
	50 to 59 years	14 (100.0)	0	
Gender	Males	71 (86.6)	11(13.4)	0.66
	Females	65 (83.3)	13 (16.7)	
Resident	Rural	48 (94.1)	3 (5.9)	0.032
	Urban	88 (80.7)	21 (19.3)	
Education	Primary	27 (60.0)	18 (40.0)	<0.001
	Secondary	53 (98.1)	1 (1.9)	
	Bachelor	30 (85.7)	5 (14.3)	
	Diploma	12 (100.0)	0	
	Masters	10 (100.0)	0	
	PhD	4 (100.0)	0	
Total		136 (85.0)	24 (15.0)	

## Data Availability

The data presented in this study is available on request from the corresponding author.

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
