# Peer review of "Caries Risk Assessment Using the Caries Management by Risk Assessment (CAMBRA) Protocol among the General Population of Sakaka, Saudi Arabia—A Cross-Sectional Study"

_ijerph, 2022, doi:10.3390/ijerph19031215_

Round 1

Reviewer 1 Report

  1. The manuscript requires comprehensive English language editing.
  2. In the materials and methods section, it is not mentioned that the radiographs of the participants were taken. However, Table 2 shows that they were conducted. This should be clarified.
  3. Line 85: the term “main researcher” is not appropriate. Replace it with “research team”.
  4. Line 83, the authors mention that the required sample size was 235. Then why only 160 participants were included in this study? This should be explained.
  5. Table 1, Number (%) column, the percentages are not correct, and they go beyond 100. For e.g. male + female (51.3 + 48.8) becomes 102. Please double-check the whole table for this.
  6. Line 104, the percentages for white spots (71.3%) and visible cavities (70%) should be mentioned separately.
  7. Figure 1, more explanation is needed as to how the participants were categorized into high and moderate caries risk groups. This can be added to the caption of Fig.1.
  8. Discussion, line 128, no need to define CAMBRA here again. You already did it in the introduction section. Just use the abbreviation.
  9. Line 137, ref. 17 is not a related study as the participants included were diabetic. The authors should include a relevant study here.
  10. Line 146, change white lesions to white spot lesions.
  11. Line 153, were these risk factors exactly prevalent at 73% in all these studies? If not, remove ‘73%’.
  12. Line 168, only use ref. 23 here (which is a clinical trial).
  13. Line 174-175, “there are some studies…..”, but for this, the authors only mention one reference (ref. 26). Either change “studies” to “study” or add more references here.
  14. There are multiple small paragraphs in the discussion section, related paragraphs should be combined.
  15. References are not according to the format of the journal. Please double-check.

Author Response

Response to comments of Reviewer 1

  1. Comment: The manuscript requires comprehensive English language editing.

Response: Manuscript was revised, and English editing was done as suggested.

  1. Comment: In the materials and methods section, it is not mentioned that the radiographs of the participants were taken. However, Table 2 shows that they were conducted. This should be clarified.

Response: Information of radiograph is added in the method section. (Line 92)

  1. Comment: Line 85: the term “main researcher” is not appropriate. Replace it with “research team”.

Response:  Revised as suggested. (Line 89 and 93)

  1. Comment: Line 83, the authors mention that the required sample size was 235. Then why only 160 participants were included in this study? This should be explained.

Response: This mistake was corrected as highlighted by reviewer. (Line 86)

  1. Comment: Table 1, Number (%) column, the percentages are not correct, and they go beyond 100. For e.g. male + female (51.3 + 48.8) becomes 102. Please double-check the whole table for this.

Response: Correction has been made in Table 1 and revised as suggested. (Table 1)

  1. Comment: Line 104, the percentages for white spots (71.3%) and visible cavities (70%) should be mentioned separately.

Response: Revised and corrected as suggested. (Line 117)

  1. Comment: Figure 1, more explanation is needed as to how the participants were categorized into high and moderate caries risk groups. This can be added to the caption of Fig.1.

Response: The explanation on how the participants were categorized into low, moderate and high caries risk groups is added in the method section. (Line  94-99)

  1. Comment: Discussion, line 128, no need to define CAMBRA here again. You already did it in the introduction section. Just use the abbreviation.

Response: Removed and revised as suggested.

  1. Comment: Line 137, ref. 17 is not a related study as the participants included were diabetic. The authors should include a relevant study here.

Response: Reference is removed, and correction has been made as suggested. (Line 142)

  1. Comment: Line 146, change white lesions to white spot lesions.

Response: Revised as suggested. (Line 151 )

  1. Comment: Line 153, were these risk factors exactly prevalent at 73% in all these studies? If not, remove ‘73%’.

Response: 73% is removed. (Line 157 )

  1. Comment: Line 168, only use ref. 23 here (which is a clinical trial).

Response: Corrected and revised as suggested. (Line 171 )

  1. Comment: Line 174-175, “there are some studies…..”, but for this, the authors only mention one reference (ref. 26). Either change “studies” to “study” or add more references here.

Response: References is added as suggested. (Line 178 )

  1. Comment: There are multiple small paragraphs in the discussion section, related paragraphs should be combined.

Response: Discussion section is improved and revised as suggested.

  1. Comment: References are not according to the format of the journal. Please double-check

Response: Format of references have been changed as mentioned.

Reviewer 2 Report

Thank you for sending me the above manuscript entitled ''Caries risk assessment using....A cross sectional study'' for peer-review. The study shows originality, however, authors may benefit from the below feedback:

-Abstract needs to be structured into Background, Methodology, Results and Conclusion. Otherwise the information all over the place and do not flow smoothly. For example, mentioning the sample size should not be after reporting the statistical tests.

-The authors need to explain and justify why not using the Caries Risk assessment tool? is it because this form is limited with children aged 6 yrs and below only? you need to make it clear.

-The Discussion section should start with reporting the aim of the study with key findings, rather than giving another definition for dental caries.

-The study will benefit from English editing.

-Was there a pilot study? this needs to be reported in  the Methodology.

-Was there power calculation of the study sample size? this needs to be reported in the Methodology.

-The study place where it has been conducted needs to be reported clearly in the beginning of the Methodology. 

-Was there any ethical approval needed? this needs to be mentioned in the beginning of the Methodology with the reference number.

-There is a benefit of mentioning the three categories of the form in the methodology to save the reader the time searching of what is the form all about?

Author Response

Response to comments of Reviewer 2

  1. Comment: Thank you for sending me the above manuscript entitled ''Caries risk assessment using....A cross sectional study'' for peer-review. The study shows originality, however, authors may benefit from the below feedback:

Response:  Thank you and noted.

  1. Comment:-Abstract needs to be structured into Background, Methodology, Results and Conclusion. Otherwise the information all over the place and do not flow smoothly. For example, mentioning the sample size should not be after reporting the statistical tests.

Response: Abstract is revised and structed into different section (Background, Method, Results and Conclusion) as suggested.

  1. Comment: -The authors need to explain and justify why not using the Caries Risk assessment tool? is it because this form is limited with children aged 6 yrs and below only? you need to make it clear.

Response: Justification for using CAMBRA in this study is added in the introduction section (Line 66-68).

  1. Comment: -The Discussion section should start with reporting the aim of the study with key findings, rather than giving another definition for dental caries.

Response: Discussion is revised as suggested. (Line 135)

  1. Comment: -The study will benefit from English editing.

Response: English editing was performed, and manuscript is revised as suggested.

  1. Comment: -Was there a pilot study? this needs to be reported in  the Methodology.

Response: Information of pilot study is added in the method section as advised. (Line 99-101)

  1. Comment: -Was there power calculation of the study sample size? this needs to be reported in the Methodology.

Response: The sample size calculation information is mentioned in method section and power calculation was not calculated. (Line 83-86)

  1. Comment: -The study place where it has been conducted needs to be reported clearly in the beginning of the Methodology. 

Response: The study place is added in the beginning of the method section as suggested. (Line 75-76 )

  1. Comment: -Was there any ethical approval needed? this needs to be mentioned in the beginning of the Methodology with the reference number.

Response: Ethical approval information with code is added in method section as suggested. (Line 86-87)

  1. Comment: -There is a benefit of mentioning the three categories of the form in the methodology to save the reader the time searching of what is the form all about?

            Response: The explanation on how the participants were categorized into low, moderate and high caries risk groups is added in the method section as advised. (Line 94-99)
